

# Global mapping of lake-terminating glaciers

Jakob Steiner[1,2], William Armstrong[3,*], Will Kochtitzky[4,*], Robert McNabb[5,*], Rodrigo Aguayo[6], Tobias Bolch[7], Fabien Maussion[8], Vibhor Agarwal[9,**], Iestyn Barr[10], Nathaniel R. Baurley[11], Mike Cloutier[12], Katelyn DeWater[4], Frank Donachie[13], Yoann Drocourt[14], Siddhi Garg[15], Gunjan Joshi[16], Byron Guzman[17], Stanislav Kutuzov[18,19], Thomas Loriaux[20], Caleb Mathias[21], Brian Menounos[21], Evan Miles[22,23,24], Aleksandra Osika[25], Kaleigh Potter[4], Adina Racoviteanu[26], Brianna Rick[27], Miles Sterner[12], Guy D. Tallentire[28], Levan Tielidze[29,30], Rebecca White[31], Kunpeng Wu[7,32], and Whyjay Zheng[33]

[1]Institute of Geography and Regional Studies, University of Graz, Austria
[2]Himalayan University Consortium, Lalitpur, Nepal
[3]Department of Geological and Environmental Sciences, Appalachian State University, Boone, NC, USA
[4]School of Marine and Environmental Programs, University of New England, Biddeford, Maine, USA
[5]School of Geography and Environmental Sciences, Ulster University, Coleraine, UK
[6]Department of Water and Climate, Vrije Universiteit Brussel, Brussels, Belgium
[7]Institute of Geodesy, Graz University of Technology, Graz, Austria
[8]School of Geographical Sciences, University of Bristol, UK
[9]Emporia State University, Kansas, USA
[10]Department of Natural Sciences, Manchester Metropolitan University, UK
[11]School of Geography and Environmental Science, University of Southampton, UK
[12]Polar Geospatial Center, University of Minnesota, St. Paul, USA
[13]Department of Geography, University of Sheffield, UK
[14]ACRI-ST, Sophia-Antipolis, France
[15]University of Dayton, Ohio, USA
[16]Department of Information Services and Computing, Helmholtz-Zentrum Dresden-Rossendorf, Dresden, Germany
[17]Universidad Austral de Chile, Valdivia, Chile
[18]School of Earth Sciences, Ohio State University, Columbus, USA
[19]Byrd Polar and Climate Research Center, Columbus, USA
[20]Universidad de Santiago de Chile, Santiago, Chile
[21]Department of Geography, Earth, and Environmental Sciences, University of Northern British Columbia, Prince George, British Columbia, Canada
[22]Department of Geography, University of Zurich, Zurich, Switzerland
[23]Department of Geography, University of Fribourg, Fribourg, Switzerland
[24]Swiss Federal Institute for Forest, Snow, and Landscape Research, Birmensdorf, Switzerland
[25]Institute of Earth Sciences, University of Silesia in Katowice, Katowice, Poland
[26]Université Grenoble Alpes, CNRS, IRD, IGE – Saint Martin d'Hères, France
[27]RedCastle Resources, Salt Lake City, Utah, USA
[28]Department of Geography and Environment, Loughborough University, Loughborough, UK
[29]Securing Antarctica's Environmental Future, School of Earth, Atmosphere and Environment, Monash University, Clayton, Australia
[30]School of Natural Sciences and Medicine, Ilia State University, Tbilisi, Georgia
[31]School of Geography, University of Leeds, Leeds, UK
[32]Institute of International Rivers and Eco-Security, Yunnan University, Kunming, China
[33]Center for Space and Remote Sensing Research, National Central University, Taoyuan, Taiwan
*These authors contributed equally to this work.





**Authors beyond this point are listed alphabetically.

**Correspondence:** Jakob Steiner (jakob.steiner@uni-graz.at), William Armstrong (armstrongwh@appstate.edu),
Will Kochtitzky (wkochtitzky@une.edu), and Robert McNabb (r.mcnabb@ulster.ac.uk)

**Abstract.** Proglacial lakes at glacier termini have received widespread attention in the literature for their role in accelerating melt, velocity and contributing to cryospheric hazards. Although global and regional inventories for both glaciers and lakes exist, lake-terminating glaciers have not been consistently identified at the global scale. Based on the most recent global glacier inventory (RGI7), which so far only identifies some marine termini but none for lakes, we present a global inventory of lake-terminating glaciers, differentiating between three classes. The dataset corresponds to the year 2000 ($\pm$ 1.5), matching to the timestamp of RGI7 outlines (2001, $\pm$ 6.2). We find that of 274,531 glaciers worldwide, 1.4 % terminate in lakes, varying between 0.5 and 6.7 % across 19 RGI regions. These glaciers account for 11.4 % of the total glacier area (0.2 to 41.8 % across regions). With multiple submitted flags available for 1260 individual glaciers, we find mapping conflicts to be low (6.7 %). The lake termini data set is available at https://doi.org/10.5281/zenodo.15524733 (Steiner et al., 2025) as well as at https://github.com/GLIMS-RGI/lake_terminating). This dataset is integrated into the forthcoming update to the RGI, v7.1.

## 1 Introduction

Lakes at glacier termini are of fundamental importance for glacier evolution, as they can increase glacier mass loss through calving and subaqueous melt (collectively referred to as frontal ablation). They also contribute to glacier velocity, and drive dynamic thinning (Boyce et al., 2007; Tsutaki et al., 2019; Trüssel et al., 2013; Larsen et al., 2015; King et al., 2019; Pronk et al., 2021; Main et al., 2023; Minowa et al., 2023; Baurley et al., 2020). Modeling has shown that the presence of a proglacial lake can accelerate grounding line recession and ice flow velocity by more than 80 % (Sutherland et al., 2020). Field measurements have shown how varying temperature, turbidity gradients, and lake circulation facilitate differential backwasting at the ice-lake interface (Röhl, 2003, 2006; Sugiyama et al., 2016) and conversely how subglacial discharge impacts the stratigraphy of proglacial lakes (Sugiyama et al., 2021). Calving is an important driver of glacier mass loss (Warren and Aniya, 1999), but remains poorly constrained with scattered observations (Warren et al., 2001; Sakai et al., 2009; Watson et al., 2020). Minowa et al. (2021) found that lake-terminating glaciers in Patagonia collectively lost $\sim 24$ Gt a$^{-1}$ through frontal ablation on average over 2000 - 2019. Knowledge about lake-terminating glaciers is also important for geodetic mass balance estimations. Zhang et al. (2023) found that the mass loss of lake-terminating glaciers in the Himalaya was underestimated by 6.5 % by existing geodetic studies, which fail to account for lake water replacing lost ice mass.

These regional assessments and case studies remain difficult to upscale, as no comprehensive global assessment of lake-terminating status exists. There are many regional lake inventories that indicate the presence of proglacial lakes (Table 1, e.g., Chen et al., 2021; How et al., 2021; Mölg et al., 2021; Rick et al., 2022; Wood et al., 2021). At the global scale, the two existing datasets (Shugar et al., 2020; Zhang et al., 2024) differ vastly in the assessment of the presence of glacial lakes, with estimates for the total number ($n$) and volume ($V$) varying by a factor of $\sim 2$ ($n = 71508$ and $V = 1280.6\pm354.1$ km$^3$ for 2020





in Zhang et al. (2024); $n$ = 14394 and $V$ = 156.5 km$^3$ in 2018 in Shugar et al. (2020)). All these inventories include lakes that are near glaciers (i.e. proglacial lakes), but most do not distinguish lakes in direct contact with glacier ice. The matter is further complicated by the rapid change in glacier cover, necessitating a lake inventory that temporally matches existing glacier outlines. A comprehensive, regularly updated global glacier inventory with a timestamp close to 2000 exists (Randolph Glacier Inventory version 7.0, RGI Consortium, 2023, hereinafter referred to as RGI7), though information about land-, marine- and lake-terminating glaciers has yet to be included at the global scale.

The aim of this study is therefore to identify lake-terminating glaciers in RGI7, using existing lake inventories as a guideline. Temporally, we focus as much as possible on the target year of ∼2000 which is consistent with the RGI7 glacier outlines.

## 2 Methods and Data

We used RGI7 (RGI Consortium, 2023) as a baseline for glaciers to be investigated. We determined that a manual approach to identify and classify lake-terminating glaciers was necessary due to the complex morphology of glacier termini, inconsistencies in the RGI7 outlines, and a lack of any globally standardized glacier lake dataset. To assemble the author team, distribute workload and ensure a community-based approach utilizing regional knowledge of the various experts, we put out an open call through the Cryolist listserv (https://lists.cryolist.org/mailman/listinfo/cryolist), from which we received more than 63 contributions from 29 individual contributors across all 19 RGI regions.

Where available, we relied on existing lake inventories for each region (Table 1) that closely matched the 2000 baseline year of RGI7. To aid this effort, we developed a Python and equivalent R script (published with the dataset, see Section 4) that used a lake inventory to produce a limited subset of RGI glaciers that should be manually evaluated for lake-terminating status. The general workflow implemented in this script is to (a) create a geodatabase of terminus locations using the *term_lat* and *term_lon* fields from RGI7; (b) buffer the terminus positions by 1 km; and (c) spatially join the existing lake dataset and buffered RGI terminus positions. For High Mountain Asia (RGI regions 13-15), the 1 km buffer was not sufficient due to erroneous lake data, with existing inventories missing a large number of smaller lakes. Because of this, mapping in this case was largely done manually, like for regions without existing inventories.

Where no inventory was available, or available inventories were too far removed temporally from 2000 (>10 years), we only relied on satellite imagery to manually identify lakes larger than ∼0.01 km$^2$ adjacent to glaciers. In most cases, the available satellite imagery was Landsat 5-7 TM and ETM+ images from ca. 2000 (±2 years). For a small portion of the Canadian Arctic and Greenland Periphery, where the latitude is outside of Landsat orbital coverage (approx. >82.7°N; RGI regions 03 and 05), we used ASTER imagery instead.



**Table 1.** Lake inventories used for the individual RGI regions. When no inventory was available, classification was performed using Landsat and ASTER imagery from ca. 2000.

| RGI Region | No | Reference | spatial coverage | temporal coverage |
|---|---|---|---|---|
| Alaska | 1 | Rick et al. (2022) | Complete | 1997 - 2001 |
| Western Canada and US | 2 | Pekel et al. (2016) | Complete | mosaic 1999 - 2022 |
| Arctic Canada N | 3 | NA | - | - |
| Arctic Canada S | 4 | NA | - | - |
| Greenland Periphery | 5 | How et al. (2021) | Complete | 2017 |
| Iceland | 6 | NA | - | - |
| Svalbard and Jan Mayen | 7 | Wieczorek et al. (2023) | Svalbard | 1936 - 1938, 1990, 2008 - 2012, 2020 |
| Scandinavia | 8 | Andreassen et al. (2022) | Norway | 2014, 2018 - 2019 |
| Russian Arctic | 9 | Moholdt et al. (2012) | Complete | 2000 - 2010 (IPY-SPIRIT images) |
|  |  | Carr et al. (2017) |  |  |
| North Asia | 10 | NA | - | - |
| Central Europe | 11 | Buckel et al. (2018) | Austrian Alps | 2009 - 2015 |
|  |  | Mölg et al. (2021) | Swiss Alps | 1980s, 2006, 2016 |
| Caucasus | 12 | NA | - | - |
| Central Asia | 13 | Wang et al. (2020) | Complete | 1990, 2018 |
|  |  | Chen et al. (2021) |  | 2008 - 2017 |
| South Asia West | 14 | Wang et al. (2020) | Complete | 1990, 2018 |
|  |  | Chen et al. (2021) |  | 2008 - 2017 |
| South Asia East | 15 | Wang et al. (2020) | Complete | 1990, 2018 |
|  |  | Chen et al. (2021) |  | 2008 - 2017 |
| Low Latitudes | 16 | Wood et al. (2021) | Peruvian Andes | 2019 |
| Southern Andes | 17 | Loriaux and Casassa (2013) | Central and Patagonian Andes | 1987, 2001, 2011 |
|  |  | Wilson et al. (2018) |  | 1986, 2000, 2016 |
| New Zealand | 18 | Carrivick et al. (2022) | Southern Alps | 1990 - 2020 |
| Antarctic & Subantarctic | 19 | NA | - | - |

In all regions, whether or not lake inventories were available, we asked contributors to manually check and revise the collection of possible lake-terminating glaciers and assign lake-terminating relevance levels (stored as *lake_level* in the database), based on the following general criteria and their expert judgment:

**Lake-terminating level 1** (Figure 1): The glacier is in direct contact with a lake that spans at least 50 % of the terminal perimeter (planimetric length of the glacier terminus), based on a visual assessment. Glaciers in this lake-terminating level





have a lake that is large enough relative to the glacier width/terminal perimeter to have a visible impact on the glacier. Visual indicators of this category may include a calving front, crevasses, and/or icebergs.

**Lake-terminating level 2** (Figure 2): The glacier is in direct contact with a lake that spans a smaller part (clearly less than 50 % but more than 10 % by visual assessment) of the terminus, or with one or more disjointed lakes that occur along the glacier margin. Potential indicators of this category included a calving front, crevasses, and/or icebergs, but are less certain than in Level 1 cases.

**Lake-terminating level 3** (Figure 3): The glacier is in contact with one or more small lakes (area >0.01 km$^2$) that collectively are in contact with <10 % of the terminal perimeter. Glaciers in this lake-terminating level do not have visible impacts due to the lake except along a very limited (<10 %) portion of the terminus. Cases that are unclear but likely are also included in this category. Additionally, we included cases where glaciers have multiple or unclear termini (e.g. ice cap margins) in this category, if expert judgment considered the adjacent lakes not relevant for glacier dynamics. In the RGI v7.1 integration, we consider these these termini to be land-terminating, given the likely minor role these lakes play for potential dynamics or melt. However, the potential for future development into larger water bodies warrants inclusion of this category.

**Not lake-terminating, level 0** (Figure 4): The glacier is not lake-terminating; its terminus is almost exclusively in contact with land or the ocean, or has a terminal water body that is so small as to seem inconsequential. We did not consider glaciers with supraglacial lakes that have not amalgamated to form one lake spanning the majority of the glacier's terminus to be lake-terminating. Similarly, we did not consider glaciers with proglacial water bodies smaller than 0.01 km$^2$ to be lake-terminating.

These criteria require a proper identification of glacier termini from the entire glacier outline. For unclear termini found at ice caps and downstream-branching glaciers, it is necessary to develop a strategy to locate the relevant portion of glacier edge for the lake level assessment. Although the terminus is defined as the lowest end of a glacier (Cogley et al., 2011), here we used a more practical approach relying on the globally available satellite images. We identified the lowest end of a glacier based on its topography and flow direction as evident from images. This task must rely on the contributor's best judgment on the perceived topography, but only by doing so can we locate the terminus region and apply the criteria for the lake-level classification. For example, in the Canadian Arctic (RGI regions 03–04), some ice-marginal lakes are not in the terminus zone but show visible influence on glacier dynamics at scale, such as extensive calving. In this case, we assigned the glacier as level 3, indicating a land-terminating glacier with potential future development driven by lake-ice interactions.

Additionally, we considered streams cutting across termini as level 2 where expert judgment considered them to have considerable impact on ice melt, a feature only encountered in Alaska. Supraglacial lakes (lakes forming entirely on top of glacier ice) were not considered unless they coalesced into large water bodies that span the majority of the terminus.

In four regions (03, Arctic Canada North; 07, Svalbard and Jan Mayen; 17, Southern Andes; 19, Subantarctic and Antarctic Islands), we identified previously unidentified marine-terminating glaciers in the course of identifying lake-terminating

**Figure 1.** Examples of lake-terminating level 1 glacier termini. Background images are Landsat 7 ETM+ false-color composites (bands 5, 4, 3). RGI7 outlines are shown in red, while lake outlines are shown in white. (a) Alsek Glacier (RGI2000-v7.0-G-01-16980) in Alaska (region 01). Landsat image acquired 2000-08-10. (b) Knik Glacier (RGI2000-v7.0-G-01-10684) in Alaska (region 01). Landsat image acquired 1999-07-31. (c) Skilak Glacier (RGI2000-v7.0-G-01-08014) in Alaska (region 01). Landsat image acquired 2000-08-09. (d) Lhotse Shar Glacier (RGI2000-v7.0-G-15-06763) in South Asia East (region 15). Landsat image acquired 2000-10-30.

**Figure 2.** Examples of lake-terminating level 2 glacier termini. Background images are Landsat 7 ETM+ false-color composites (bands 5, 4, 3). RGI7 outlines are shown in red, while lake outlines are shown in white. (a) Malaspina (Sít' Tlein) Glacier (RGI2000-v7.0-G-01-15261) in Alaska (region 01). Landsat image acquired 2000-09-09. (b) Russel Glacier (RGI2000-v7.0-G-01-16437) in Alaska (region 01). Landsat image acquired 2000-08-31. (c) Inilchek Glacier (RGI2000-v7.0-G-13-28434) in Central Asia (region 13). Landsat image acquired 2000-09-13. The main lake is not in the regional lake inventory but is visible in satellite imagery (and a known lake that is present most of the time but repeatedly drains as a glacial lake outburst flood). (d) Middle Fork Glacier (RGI2000-v7.0-G-01-06125) in Alaska (region 01). Landsat image acquired 1999-08-11.

**Figure 3.** Examples of lake-terminating level 3 glacier termini. Background images are Landsat 7 ETM+ false-color composites (bands 5, 4, 3). RGI7 outlines are shown in red, while lake outlines are shown in white. (a) Chisana Glacier (RGI2000-v7.0-G-01-05589) in Alaska (region 01). Landsat image acquired 1999-08-11. (b) Unnamed glacier (RGI2000-v7.0-G-01-04360) in Alaska (region 01). Landsat image acquired 2000-08-16. (c) Unnamed glacier (RGI2000-v7.0-G-01-16871) in Alaska (region 01). Landsat image acquired 2000-08-10. (d) Ghiaccio del Cavagn (RGI2000-v7.0-G-11-02473) in Central Europe (region 11). Landsat image acquired 2000-08-28.





**Figure 4.** Examples of lake-terminating level 0 glacier termini. Background images are Landsat 7 ETM+ false-color composites (bands 5, 4, 3). RGI7 outlines are shown in red, while lake outlines are shown in white. (a) Eklutna Glacier (RGI2000-v7.0-G-01-10928) in Alaska. Landsat image acquired 1999-07-31. (b) Unnamed glacier (RGI2000-v7.0-G-01-11048) in Alaska (region 01). Landsat image acquired 1999-07-31. (c) Harris Glacier (RGI2000-v7.0-G-01-08628) in Alaska (region 01). Landsat image acquired 2000-08-09. (d) Hispar Glacier, with numerous supraglacial ponds (RGI2000-v7.0-G-14-21670) in South Asia West (region 14). Landsat image acquired 2000-09-11.





glaciers using available Landsat imagery. In two of these regions (03 and 19), we also identified glaciers terminating in landfast ice or ice shelves. In region 03, we did this identification using the available Landsat imagery. For region 19, we compared the *TermType* attributes from version 6.0 of the RGI, high resolution coastline polylines from the Scientific Committee on Antarctic Research Antarctic Digital Database (Gerrish et al., 2024), and available Landsat imagery. To keep these glaciers distinct and aid in using this database to update the attributes of later RGI releases, we assigned marine-terminating glaciers a *lake_level* of 99, and shelf-terminating glaciers a *lake_level* of 98. For the purposes of estimating the total number and area of lake-terminating glaciers in each region, we treat these as having a *lake_level* of 0 (i.e., not lake-terminating).

We acknowledge that neither the RGI7 nor any lake database is perfect, including the dataset presented here. Given the complexity of the glacier-lake interface and the quality of imagery available, there can be uncertainty when judging whether a lake is actually ice-adjacent at the time of the inventory or simply in very close proximity. While this is less relevant for questions of hydrology, or even the potential lake hazard, it is more relevant in considering the influence on ice dynamics and upstream glacier mass loss. Cases where contributors identified a very close lake but had doubts over its impact were classified as level 3. For a total of 1041 glaciers more than one submission existed, which we used to assess the agreement of classification across contributors.

Including all lake level categories described above in this database, even those with lower lake levels, will support future efforts where lakes could expand as glaciers continue to retreat (e.g., Furian et al., 2021) and allowed contributors to make fewer subjective judgment calls on lake-terminating status. Future versions of this database, especially when they target a different year, will benefit from this level 3 classification as changes in the terminus position may change the lake properties and could alter the connection level.

The data are stored and published as a comma-separated variable (CSV) file for each region, following a simple structure (Table 2). The fields of the CSV file include the RGI ID (*rgi_id*), the assigned lake level (*lake_level*), the ID of the satellite image used (*image_id*) and its respective date (*image_date*), the DOI of the lake inventory used (*inventory_doi*), and an identifier for the contributor(s) who classified the glacier (*contributor*). This structure allows for the lake termini product to be integrated seamlessly when processing RGI data.

Additionally, we have used this dataset to update the *term_type* attribute for RGI7 as follows, for all glaciers where the *term_type* attribute was previously not assigned (*term_type* = 9). Glaciers classified as *lake_level* 1 and 2 in this study are given a *term_type* value of 2 in RGI7 (i.e., lake-terminating). Glaciers classified with *lake_level* values of 0 or 3 are given a *term_type* value of 0 (i.e., land-terminating). Finally, any glaciers classified in this study as marine-terminating (*lake_level* = 98) or shelf-terminating (*lake_level* = 99) are given a *term_type* value of 1 or 3, respectively, in RGI7.





**Table 2.** Data structure of the lake-terminating dataset.

| rgi_id | lake_level | image_id | image_date | inventory_doi | contributor |
|---|---|---|---|---|---|
| RGI2000-v7.0-G-13-00001 | 0 | LT51540331999231RSA03 | 1999-08-19 | 10.5194/essd-13-741-2021 | 14 |

## 3  Results and Discussion

Of the 274,531 RGI7 glaciers, 1.4 % were found to be lake-terminating, with 1 % (2733) classified as *lake_level* 1, and the remaining 0.4 % (1103) as *lake_level* 2 (Table 3). 98.1 % (269,303) were not lake-terminating (*lake_level* 0 for land-terminating cases, *lake_level* 98 and 99 for marine-terminating cases), with an additional 0.5 % (1392) presenting only some adjacent water bodies, but without significant interaction with the glacier ice (*lake_level* 3). In individual regions, *lake_level* 1 glaciers make up more than 1 % of all glaciers in 9 of the 19 RGI regions, with a maximum of 3.9 % in region 8 (Scandinavia). The relevance

of lake-terminating glaciers increases when considering the area of the respective glaciers. While land-terminating glaciers (levels 0 and 3) comprise the vast majority of global glaciers by number (98.6 %), they only make up 88.5 % of the total glacier area. In contrast, while level 1 and 2 lake-terminating glaciers account for 1.4 % of glaciers by number, they comprise 11.5 % by area (Table 3), reaching more than 20 % in regions 6 (Iceland, 33.1%) and 17 (Southern Andes, 41.8%). This prevalence of lake termini on larger glaciers is apparent from their relative frequency by area (Figure 5a), with the median size of all glaciers

(0.17 km$^2$, $\sigma$=38.42 km$^2$), considerably smaller than for those lake-terminating (0.96 km$^2$, $\sigma$=110.03 km$^2$). Other potential explanatory variables like glacier slope are not further explored here.

The difference between lake-terminating glacier number versus area becomes more apparent when comparing different regions (Table 3, Figure 5 and 6). In the Russian Arctic (Region 09), the Caucasus (Region 12) and all three regions of High

Mountain Asia (13 - 15), both the relative number and area of glaciers with lake termini (level 1 and 2) is below 5 %. In Alaska (region 01), Iceland (region 06) and the Southern Andes (region 17) on the other hand, the fraction of lake termini is low (0.7, 2.5 and 1.9 %, respectively), but the associated glaciers respectively make up 26.7, 33.1 and 41.8 % of the regional glacier area. These regions contain an abundance of large proglacial lakes, with level 1 termini alone respectively making up 16.7, 14.4 and 39.6 % of the regional glacier area. In Scandinavia, 3.9 % of all glaciers have level 1 termini (11.9 % by area). In Central Asia

(13), South Asia West (14) and South Asia East (15), these relative numbers are much lower (0.7, 0.8 and 1.9 %, respectively). In Central Asia and South Asia East, however, level 1 lake-terminating glaciers still constitute 2.7 and 3 % of the total area, while in South Asia West, their area is much smaller (0.6 %).

Across RGI regions, "hotspots" of lake-terminating glaciers vary depending on the metric of interest. Considering the pro-

portion of global lake-terminating glaciers by number, High Mountain Asia (RGI 13-15) alone accounts for over one-third of the world's glaciers terminating in lakes (Figure 7a). The Southern Andes (RGI 17) are the next largest contributor at 15 %,



**Figure 5.** (a) Number of glaciers per size category (log), for all glaciers in RGI7, and for each lake-terminating level. (b) Fraction of glaciers per size category (log), for all glaciers in RGI7, and for each lake-terminating level. (c) Relative area of lake-terminating glaciers per RGI region. Bars are labeled with the total area (in km$^2$) of lake-terminating glaciers for each region.





**Table 3.** Number $n$, area $A$ (in 1000 km²), and percent number / percent area for glaciers in each RGI region, by lake-terminating level. See Table 1 for region names and corresponding region number.

| Region | Total glaciers | | Level 0 | | | | Level 1 | | | | Level 2 | | | | Level 3 | | | |
|---|---|---|---|---|---|---|---|---|---|---|---|---|---|---|---|---|---|---|
| | $n$ | $A$ | $n$ | % | $A$ | % | $n$ | % | $A$ | % | $n$ | % | $A$ | % | $n$ | % | $A$ | % |
| 01 | 27509 | 86.7 | 27263 | 99.1 | 58.4 | 67.4 | 144 | 0.5 | 14.5 | 16.7 | 43 | 0.2 | 8.7 | 10.0 | 59 | 0.2 | 5.1 | 5.9 |
| 02 | 18730 | 14.5 | 18241 | 97.4 | 12.6 | 87.1 | 204 | 1.1 | 0.7 | 4.9 | 148 | 0.8 | 0.8 | 5.4 | 137 | 0.7 | 0.4 | 2.6 |
| 03 | 5216 | 105.4 | 4929 | 94.5 | 82.9 | 78.6 | 98 | 1.9 | 5.9 | 5.6 | 86 | 1.6 | 6.7 | 6.4 | 103 | 2.0 | 9.9 | 9.4 |
| 04 | 11009 | 40.5 | 10758 | 97.7 | 28.6 | 70.5 | 101 | 0.9 | 2.7 | 6.6 | 83 | 0.8 | 7.3 | 18.1 | 67 | 0.6 | 1.9 | 4.7 |
| 05 | 19994 | 90.5 | 19528 | 97.7 | 76.0 | 84.0 | 173 | 0.9 | 5.7 | 6.3 | 121 | 0.6 | 2.9 | 3.2 | 172 | 0.9 | 5.8 | 6.4 |
| 06 | 568 | 11.1 | 540 | 95.1 | 6.4 | 58.0 | 5 | 0.9 | 1.6 | 14.4 | 9 | 1.6 | 2.1 | 18.7 | 14 | 2.5 | 1.0 | 8.8 |
| 07 | 1666 | 34.0 | 1569 | 94.2 | 29.7 | 87.5 | 24 | 1.4 | 1.2 | 3.6 | 36 | 2.2 | 0.8 | 2.3 | 37 | 2.2 | 2.2 | 6.5 |
| 08 | 3410 | 2.9 | 3146 | 92.3 | 2.4 | 80.8 | 133 | 3.9 | 0.4 | 11.9 | 94 | 2.8 | 0.1 | 4.3 | 37 | 1.1 | 0.1 | 2.9 |
| 09 | 1069 | 51.6 | 1021 | 95.5 | 49.1 | 95.1 | 15 | 1.4 | 1.0 | 1.9 | 17 | 1.6 | 0.7 | 1.4 | 16 | 1.5 | 0.8 | 1.6 |
| 10 | 7155 | 2.6 | 7051 | 98.5 | 2.6 | 98.0 | 32 | 0.4 | 0.0 | 0.7 | 37 | 0.5 | 0.0 | 0.9 | 35 | 0.5 | 0.0 | 0.4 |
| 11 | 4079 | 2.1 | 3969 | 97.3 | 1.9 | 89.6 | 46 | 1.1 | 0.1 | 4.4 | 20 | 0.5 | 0.0 | 1.9 | 44 | 1.1 | 0.1 | 4.1 |
| 12 | 2275 | 1.4 | 2250 | 98.9 | 1.4 | 96.4 | 3 | 0.1 | 0.0 | 0.2 | 8 | 0.4 | 0.0 | 2.1 | 14 | 0.6 | 0.0 | 1.3 |
| 13 | 75613 | 50.3 | 74664 | 98.7 | 47.1 | 93.6 | 547 | 0.7 | 1.9 | 3.8 | 168 | 0.2 | 0.6 | 1.3 | 234 | 0.3 | 0.7 | 1.4 |
| 14 | 37562 | 33.1 | 37179 | 99.0 | 31.1 | 93.9 | 305 | 0.8 | 0.5 | 1.5 | 32 | 0.1 | 0.0 | 0.1 | 46 | 0.1 | 1.5 | 4.5 |
| 15 | 18587 | 16.0 | 18007 | 96.9 | 14.2 | 88.2 | 339 | 1.8 | 1.3 | 8.1 | 81 | 0.4 | 0.2 | 1.3 | 160 | 0.9 | 0.4 | 2.4 |
| 16 | 3695 | 1.9 | 3601 | 97.5 | 1.7 | 88.6 | 50 | 1.4 | 0.1 | 5.5 | 22 | 0.6 | 0.0 | 2.3 | 22 | 0.6 | 0.1 | 3.7 |
| 17 | 30634 | 27.7 | 29882 | 97.5 | 15.5 | 56.0 | 496 | 1.6 | 11.0 | 39.6 | 79 | 0.3 | 0.6 | 2.2 | 177 | 0.6 | 0.6 | 2.2 |
| 18 | 3018 | 0.9 | 2995 | 99.2 | 0.7 | 80.0 | 7 | 0.2 | 0.1 | 16.8 | 8 | 0.3 | 0.0 | 1.9 | 8 | 0.3 | 0.0 | 1.3 |
| 19 | 2742 | 133.4 | 2710 | 98.8 | 133.0 | 99.7 | 11 | 0.4 | 0.1 | 0.1 | 11 | 0.4 | 0.2 | 0.1 | 10 | 0.4 | 0.1 | 0.1 |
| **Global** | **274531** | **706.7** | **269303** | **98.1** | **595.3** | **84.2** | **2733** | **1.0** | **48.8** | **6.9** | **1103** | **0.4** | **32.0** | **4.5** | **1392** | **0.5** | **30.7** | **4.3** |

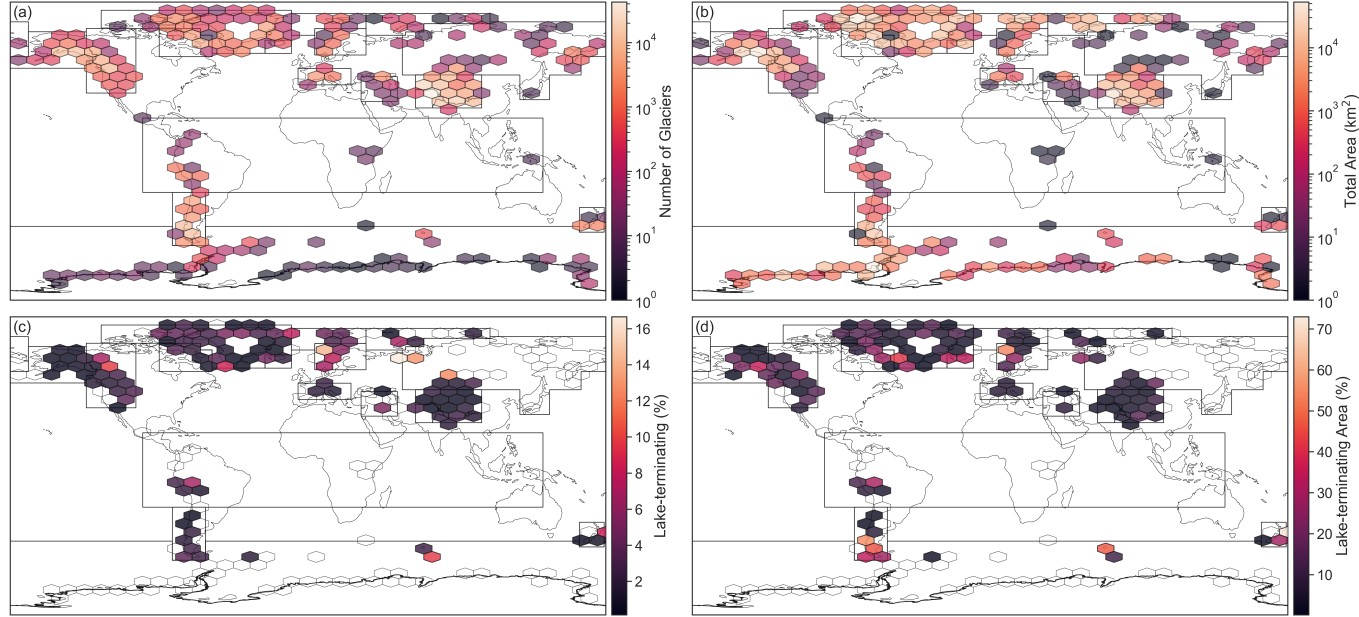

**Figure 6.** Statistics aggregated to 10° hexagonal tiles. (a) Total number of glaciers according to RGI7. (b) Total area of glaciers according to RGI7 [$km^2$]. (c) Relative number of glacier termini classified as level 1 and level 2 [%]. (d) Relative area of glaciers with termini classified as level 1 and level 2 [%].

with every other RGI region individually containing <10 % of the world's lake-terminating glaciers. Using the global proportion of lake-terminating glacier area as a metric produces an entirely different picture (Figure 7b), with Alaska (RGI 01) alone accounting for $\sim 30\%$ of the world's lake-terminating glacier area. Arctic Canada (RGI 03-04) holds another $\sim 30\%$, the Southern Andes (RGI 17) hosts $\sim 15\%$, and all other regions combined contain the remaining just over $\sim 25\%$ of global lake-terminating glacier area.

The prevalence of lake-terminating glaciers in North America may in part reflect the region's geologic history, with the relatively low regional slope of the Canadian Shield, extensive past continental glaciation, large modern glaciers and fjord like topography emanating from an interplay of glacial erosion and resistant underlying rock. Low regional slope and large glacier area are both associated with the growth of proglacial lakes in Alaska (Field et al., 2021), potentially due to these quantities being associated with the more likely existence of large subglacial overdeepenings capable of hosting substantial proglacial lakes. In contrast, the high proportion of High Mountain Asia lake-terminating glaciers by number but small proportion by area may reflect the region's high average slope, less conducive to the formation of large proglacial lakes. In addition, the stability of the dam impounding the proglacial lake may partly explain regional variations in lake-terminating glacier abundance. The failure of a moraine dam impounding a proglacial lake is a common occurrence in High Mountain Asia, while it is rare in Alaska (Rick et al., 2023). The greater stability of moraine dams in Alaska, which again likely reflects the low average slope





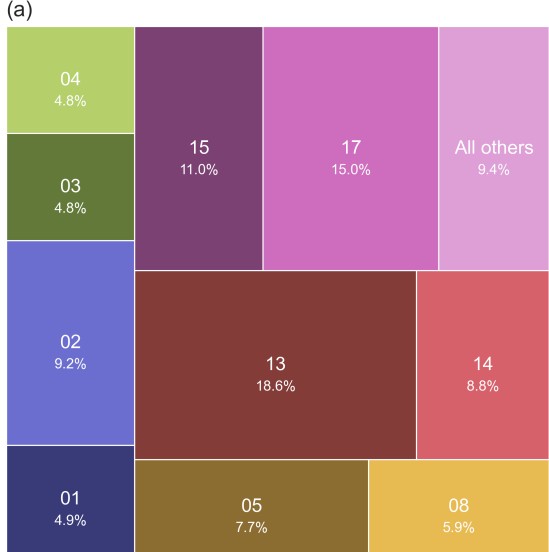
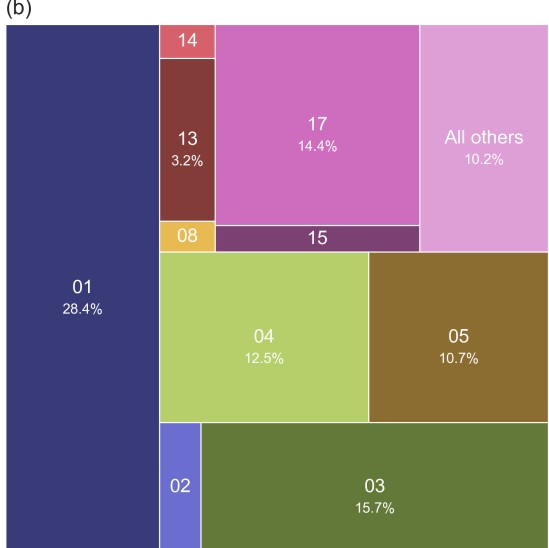

**Figure 7.** Proportion of global lake-terminating glaciers in each RGI region, shown by (a) proportion of global number lake-terminating glaciers, and (b) proportion of global lake-terminating glacier area. RGI region numbers, with the percent attributable to that region shown within the chart. Lake-terminating glaciers are here defined as those with Level 1 or Level 2 termini. Regions with $< 2\%$ of the global total are combined into the "all others" category.

at glacier termini at regional scale, may allow for the persistence of terminal lakes that are less common in other regions.

While hazards associated with glacial lakes (i.e. glacial lake outburst floods) do often also occur from lakes without direct ice contact, hence making a much larger number of proglacial lakes potentially dangerous, drivers for outburst floods like glacier calving, only play a role for glaciers with direct lake contact. Similarly, when considering the role of glacial lakes on ice dynamics, only ice contact lakes are relevant in potentially accelerating mass loss or increasing velocities. While earlier studies have focused on all potential proglacial lakes (between 14394 (or 9414 for the 1990-1999 period, comparable to the

time used for the inventory in this study Shugar et al., 2020) and 71505 (Zhang et al., 2024)), we find only 3835 glaciers that have direct lake contact. However, the results also suggest that the relative importance of these proglacial lakes with direct ice contact varies widely between the different regions.

### 3.1 Mapping uncertainty

To evaluate the congruency of mapping for glacier as well as lake outlines, we extracted time stamps for images used to create

RGI7 outlines as well as images used to check terminus type for lakes, which happened for $75.49\%$ of all flags (Figure 8). The average ($\pm 1\sigma$) image year for RGI7 is 2001 ($\pm 6.2$), while for lake flags it was 2000 ($\pm 1.5$).

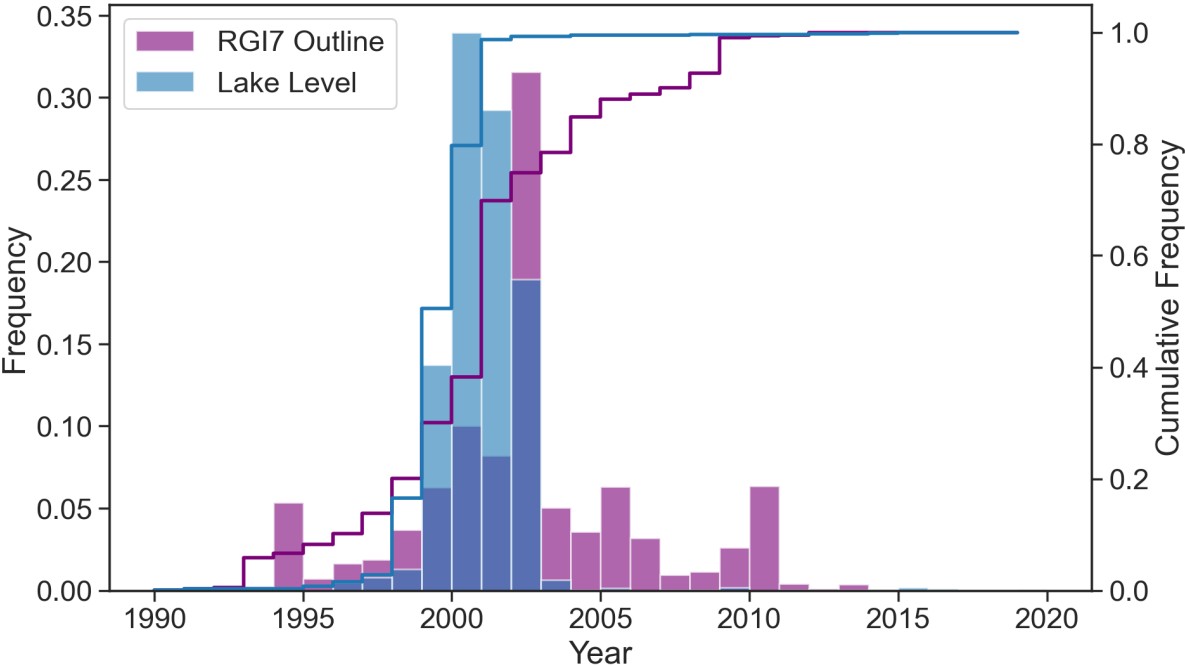

**Figure 8.** Distribution of timestamp of imagery used to generate RGI7 outlines and for classifying lake-terminating level.

Participants either submitted files that they worked on individually, or files that were first discussed and compiled before being submitted for ingestion. From the files that were submitted by individuals, a total of 45,047 glaciers across seven regions were mapped by more than one participant (16 % of all glaciers). This was generally the case when one person did initial mapping and a second revised the judgment, without necessarily storing the original choice. For 1260 glaciers at least two separate flag submissions were available. Note that these duplicate submissions were not done with the primary intent to check consistency, but were sourced from all submissions, which suggests that the operators' eye to detail was no different from any other submission. Of these, participants disagreed about the lake level classification for only 84 glaciers (6.7 %); in the majority of cases (at least 72 out of 84), these disagreements were due in part to participants using different images for the classification. In most cases (42 out of 84, 50 %), the disagreement was between level 0 and 3, which we both consider land-terminating. In 21 of these cases the final agreed lake level was 0, in 20 it was 3, once the reassessment settled for 2. The decision, whether the glacier is in contact with some or small lakes over a very limited portion of the terminus (our definition for level 3), or almost exclusively in contact with land (level 0) is understandably subjective and with the current lack of precise and consistent lake outlines matching the time stamp of glacier outlines, remains impossible to tackle with a consistent automatic approach. In 29 cases (35 %) there was disagreement between lake-terminating (1 or 2) and land-terminating (3 or 0), of which 12 were subsequently to be judged land-terminating and 17 lake-terminating. In 5 of these 29 cases, the disagreement was between lake level 2 and 3, which again provides opportunity for subjectivity as deciding whether the ice-lake interface spans more or less

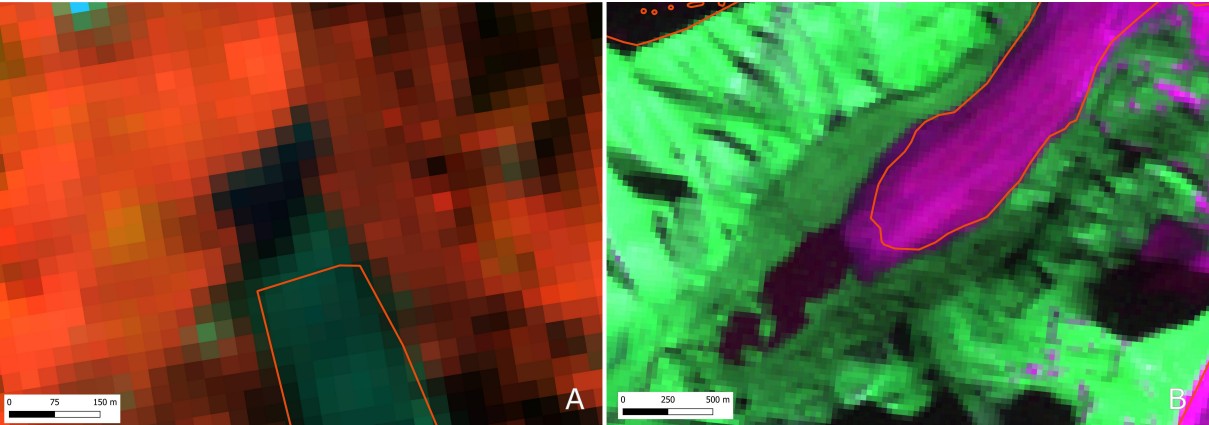

**Figure 9.** Examples of glaciers where operators disagreed on lake levels for eventual lake-terminating level 1 cases. (A) Shudder Glacier (RGI2000-v7.0-G-02-07418), mapped as level 1 and 2. (B) Ragnarbreen (RGI2000-v7.0-G-07-00470), mapped as level 1 and 0. Background images are Landsat 7 ETM+ false-color composites (bands 5, 4, 3).

than 10 % is for now impossible to quantify exactly. The remaining cases are largely subject to different imagery used, and the quality of the ice-lake interface was considerably different or lakes had appeared between one or the other, potentially possible considering the time range of imagery used (Figure 8). Often the disagreement was simply because of differing prioritization of glacier masks versus available imagery. Figure 9 shows two cases of level 1 termini, where the RGI outlines alone do not suggest a clear interface with the present lake body (and were hence marked as level 2 and 0, by one operator, respectively), while the imagery clearly suggests a substantial contact. Although in these cases the final level was stored as level 1, it suggests that in cases where the outline does not clearly provide an overlap with the lake outline there may be more erroneous misclassifications.

Where there was disagreement between participants over the classification for a glacier, the disagreement was resolved either by having a third person check the glacier and make a decision, or by having participants discuss and come to an agreement. We acknowledge, that complete agreement as for glacier outlines is likely unattainable. We suggest, however, that future availability of consistent global lake inventories with a clear time stamp and based on high resolution imagery becoming increasingly available, will allow to further increase agreement on terminus type.

## 4 Code and data availability

The data generated for this study are available at https://doi.org/10.5281/zenodo.15524733 (Steiner et al., 2025). The data includes all material documenting the process described in this manuscript. The final dataset produced is stored in the *tables* folder, including the final *.csv* files for each RGI7 region following the format of RGI7, allowing for a seamless integration (Ta-

ble 2). Furthermore, the data are available as geopackages (*.gpkg*), allowing for rapid access in a GIS environment to all glacier outlines with lake levels 1 to 3, as well as spatial points at all glacier centroids, with the respective lake level as an attribute. The folder also includes files that store all operator IDs (*contributors.csv*), the number of glaciers that have been mapped by multiple operators per region (*multiple_operators.csv*) and a summary of glaciers where conflicts arose (*mappingconflicts.csv*).

The development of the dataset can be accessed at https://github.com/GLIMS-RGI/lake_terminating. The scripts used to produce the figures and tables in this manuscript are available at https://github.com/GLIMS-RGI/lake_terminating/tree/main/essd. The data will also be integrated into the forthcoming RGI update, v7.1.

## 5 Conclusions

The importance of proglacial lakes is often noted across many glacierized mountain regions, whether it is for their potential to influence glacier dynamics, glacier mass loss, or their role in cryospheric hazards. While glacial lakes without a direct ice-water interface can potentially play an important role in mountain hydrology and limnology and as a source of glacial lake outburst floods, direct contact is necessary for the lake to have any potential influence on glacier dynamics and mass loss. In this study, we, for the first time, quantify the total number of glaciers in direct contact with a lake (3836, 1.4 % of all glaciers in RGI7) as well as those that have a potentially significant role in influencing dynamics (2733, 1 %). While these numbers are relatively low at the global scale, they rise up to 6.9 and 3.9 %, respectively, in individual regions. Glaciers with adjacent lakes tend to be larger, with glaciers with lake termini making up 11.4 % of the total glacier cover, and 6.9 % of glaciers with termini where the proglacial lake was identified to play an important role for glacier dynamics (level 1).

Mapping exercises like the one conducted here are always subject to errors as well as some level of subjectivity when judging the role lakes may play at the glacier terminus. Relying on 1260 glaciers, where at least two individual operators independently flagged the termini, we can show that only for 6.7 % of all glaciers conflicts in judgment occurred. For 50 % of these conflicts the disagreement was over different levels of land-terminating glaciers (i.e. level 0 and 3). The remaining disagreements suggest a relatively small margin of error in the dataset to correctly identify lake termini, largely subject to different imagery chosen for the analysis or the glacier masks being used as defining over underlying imagery.

We suggest that the relative importance of proglacial lakes should be further investigated, taking these regional variations as well as the relative fraction by number and area into account. While the numbers presented here suggest that proglacial lakes do likely play an important role in some regions when it comes to influencing glacier dynamics, thinning or mass loss, it is important to consider which glaciers are in fact in contact with lake water.

The present analysis does not consider the actual number of lakes in contact with an individual glacier or the length of the ice-water interface, remaining a task for future analysis. The inventory furthermore only considers the 2000 timestamp, to



align with the available glacier inventory. Future studies could use this inventory to evaluate the changing nature of termini with adjacent lakes. Investigations into the role of the presence or absence of a proglacial lake on mass loss and ice dynamics is now possible at the global scale. This would improve our understanding of the relative importance of change along the ice-water interface, irrespective of the growing number of glacial lakes.

*Author contributions.*   FM initiated the study, RA, WA, TB, WK, RM and JS conceptualized the study and coordinated the mapping effort. JS led the writing and JS and RM did analysis, RM created figures with input from JS, RA, WA and TB. All others contributed by flagging individual termini in one or more regions as well as with reviewing the manuscript.

*Competing interests.*   The authors declare no competing interests.

*Acknowledgements.*   The authors thank Alexandre Bevington and Matt Nolan for supporting the study with satellite imagery and known lake
locations. We thank Regine Hock for comments on presentation of this work that informed the text.





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
