# Peer review of "Global mapping of lake-terminating glaciers"

_Earth System Science Data, 2025_

## Author Comment (AC1)

**Response to comments on "Global mapping of lake-terminating glaciers" (essd-2025-315)**

**Reviewer 1 (Penny How)**

Thank you for providing such a careful reading of the manuscript, including the dataset and its accessibility. We also appreciate the proactive comments directly on github to further improve the product. We address each point individually below in red, identifying the resulting changes either in manuscript or the database itself.

Steiner et al. present a community-led effort to identify and categorise lake-terminating glaciers globally, which is compatible with the Randolph Glacier Inventory version 7.0 (RGI7) and intended for future integration. The dataset has been primarily generated manually through a concerted and coordinated effort from the authors. Initial categorisations have been formed from existing global and regional glacial lake inventories, drawing upon and uniting a large portion of the glacial lake mapping and monitoring efforts. Error is estimated based on comparing classifications from two operators, which reveals low mapping conflicts (6.7 %) that is indicative of a low uncertainty in the dataset.

The dataset itself is logical and clear to a large extent, as reflected in the dataset description manuscript. The dataset description paper is well written and a thorough companion to the dataset. My feedback is mainly on the dataset itself and the accompanying Github repository, with my primary focus being to ensure that the dataset is unambiguous to users in the glaciology/cryosphere research community and beyond. Github repository pull requests and issues corresponding to some of this feedback have been included here, and I have added my comments early in the review process so that a dialogue can continue on the Github repository if needed.

In all, I would recommend acceptance after these revisions. I am looking forward to seeing this dataset integrated with RGI7. Great work!

**Dataset comments**

1. The naming of the lake-terminating glacier classifications

In the dataset and throughout the manuscript, the classifications to describe the relevance of lake presence are referred to as "lake level", "lake\_level" and "lake level assessment". The term "lake level" is often used in reference to the water level of a lake, for example in remote sensing (e.g. Veh et al., 2025), modelling (e.g. Steffen et al., 2022), and in situ studies and monitoring efforts (e.g. Camassa et al., 2023).

I recommend that the naming convention is changed to something more suitable and less ambiguous. My suggestion would be "lake category" ("lake\_catgy" in the gpkg field name), with Level 0-3 renamed to Category 0-3 (and capitalised throughout the manuscript).

This is a fair comment, and we have now implemented this (i.e. changing 'level' to 'category') throughout the manuscript, in the dataset itself ("lake\_level" to "lake\_cat"), the README files, and the original contributor files.

2. Level 0 to Level 3 categories are not in sequential order

The Level 0 to Level 3 relevance classifications are not in sequential order despite their numbering convention. Specifically, no lake contact (Level 0) is followed by > 50 % lake contact (Level 1), then < 50 % lake contact (Level 2), and then < 10 % and/or ambiguous lake contact (Level 3).

The classification levels should follow the magnitude of relevance sequentially, therefore my suggestion is:

- Category 0: no lake contact
- Category 1: < 10 % and/or ambiguous lake contact
- Category 2: < 50 % lake contact
- Category 3: > 50 % lake contact

Where "Level" is replaced by "Category" in accordance with the recommendation above. The dataset, processing scripts, manuscript, repository readme, and statistical analysis should also be updated accordingly.

Thank you for the suggestion, we have followed it and this has now been implemented throughout the manuscript, the ReadMe files, the dataset itself, and the original contributor files.

**3. Ambiguous relevance classifications**

The definitions of the relevance classifications (Level 0 to Level 3) differ between the ESSD manuscript and the Github repository readme, where the repo readme explicitly describes the relevance to glacier behaviour whereas the ESSD manuscript merely infers this. I would suggest amending the Github repository readme to align with the ESSD definitions, given that it is problematic to define an explicit connection between glacier-lake coverage and the certainty/amplitude of its impact on glacier behaviour. I have made a PR with these proposed changes: https://github.com/GLIMS-RGI/lake\_terminating/pull/12.

Thank you for the suggestion. We agree that the manuscript and the repository should align and that too strong of an explicit connection may not be warranted (or necessary). We follow the PR and have merged this together with the suggested changes on naming. We have not completely aligned the text in repository and manuscript, to also preserve the nature of how this was built. Guiding documents were written at the beginning of the exercise as guidance for a large group of operators, while the manuscript reflects our collective understanding at the end of it.

Additionally, there appears to be ambiguity surrounding the criteria for each relevance classification. At various points in the manuscript, the relevance classification signifies:

- i) The portion of terminus in contact with lake (e.g. Line 70-72).
- ii) The perceived level of influence on the adjacent glacier based on visual indicators (e.g. Line 77-78)
- iii) The operator certainty of the classification/ice contact (e.g. Line 76-77, 113-115)

Therefore, the relevance classification is ambiguous as it indicates more than one criteria. In future iterations of this dataset, I propose that criteria i) and ii) should be separated from iii), with a new field denoting the operator certainty. In addition, the criteria for the relevance classification should be revised and clarified in the manuscript (Line 63-84).

Thank you for catching these inconsistencies. We have added a line before the 4 categories ('predominately resting on the relative proportion of glacier ice intersecting with lake water and the potential resulting effects of this connection'). We have also removed the ambiguous line 'Cases where contributors identified a very close lake but had doubts over its impact were classified as Category 1.', reducing the criteria to (i) and (ii). We agree that the operator uncertainty would ideally be classified in a different manner. With the difficulty of clearly framing this uncertainty across different experts, this hasn't been attempted during the compilation. We have however added this as a recommendation for potential future iterations as 'Although a consistent qualification of (un-)certainty across multiple contributing experts remains challenging, future iterations of this dataset should consider including a separate attribute that lets contributors self-assess their certainty for individual cases.'

**4. Dataset directory naming/structuring conventions**

It is difficult to locate the dataset itself in the Github/Zenodo repository alongside the data handling scripts and documentation. I propose renaming the directory from "tables" to "dataset" in order to make this clearer, and ensuring that only the finalised dataset is in the top level of the "dataset" directory (i.e. moving all un-collated operator definitions to a sub-directory). I have opened a pull request to the Github repository (https://github.com/GLIMS-RGI/lake\_terminating/pull/11) with these proposed changes.

Implemented with PR#14.

5. The Greenland periphery glacier outlines .gpkg file is missing from dataset

https://github.com/GLIMS-RGI/lake\_terminating/issues/10

Fixed with PR#14.

**Line-by-line paper comments**

I don't have many line-by-line comments, largely because the language and communication of the findings presented in the manuscript are to an excellent standard. Therefore, my line-by-line comments are largely remarks, questions and figure/table queries.

Line 19-20: I am unsure about the general statement that calving "remains poorly constrained with scattered observations", especially given that two of the three references to support this are over 15 years old. Can the statement be amended to better reflect the advances in calving modelling and integration into system models over recent years.

Without doubt the literature on calving (modelling and observations) is ample, but largely revolves around marine termini or even just ice shelves. We have now specified here that our statement pertains to calving into lakes (we think diving into general calving studies here would go too far) and have added a more recent study that has investigated this locally (with the following study by Minowa et al. also pointing to one of the few more comprehensive quantifications).

Line 28-31: I did not realise that these two global inventories differed so greatly, therefore it is good to see this reported here. Do you know why the difference is so vast? Is this a reflection of the difference in classification approaches and/or discrepancies in manual intervention?

The differences in these two inventories (or rather concern with the quality of each individually when looking at individual lakes) has indeed been a concern, if not further investigated. While we have not attempted to determine the source of discrepancies, the difference we believe lies in the different classification approaches.

Line 49: Great to see the processing script openly provided for this.

**Thank you!**

Line 58: Was there any specific reason for choosing a lake size threshold of 0.01 sq km? Was this problematic in cases where an existing inventory only contained lakes with a higher minimum size threshold (e.g. Greenland, with a minimum size of 0.05 sq km)?

The threshold was chosen based on Landsat pixels being potentially able to capture lakes of such a size but not below and numerous known lakes that interact with glaciers falling below a higher threshold like 0.05 km². We acknowledge that to some degree this is a bit of a subjective choice. For cases where inventories with higher thresholds were used, the manual checking that was employed was able to address this.

Line 79: Repetition of "these".

**Thanks, edited.**

Table 2: I would like to see the entry type (i.e. string, integer, float) for each of these fields, mainly to guide users who are importing these using R or Python. Also, a short description for each field should be added, similar to those described in Lines 124-128.

Thanks for the suggestion, we have added this to the caption now.

Line 123-127: The AutoTerm field is not defined here, in Table 2, or in the Github repository. I am guessing this is a categorisation of the level to which an external glacial lake inventory dataset was used?

This field is the modified *term\_type* field, updated after running **scripts/assign\_lake\_flag.py**. We have now added the definition/explanation of this field at Line XX of the manuscript, and to the README file in the repository.

Line 149: "...(Table 3, Figure 5 and 6)." >> "...(Table 3, Figure 5 and 6)."

**Corrected.**

Table 3: I think the region name should be included here, if possible, rather than having to refer to Table 1.

While in principle we agree that this would ease the readability, we are already maxing the page here quite far and adding the full names would in our view use unnecessary space, which we need here for the numbers. We would therefore prefer to simply stick to the IDs.

Line 194-216: An additional table would neatly summarise and compliment these findings (i.e. "Table 4. Statistics from independent flag submissions of glacier classifications")

Thank you for this suggestion. We have now added a confusion matrix of the categories assigned by multiple reviewers (Table 4).

Line 218-220: Are these discussions openly available, for instance through issue postings on the Github repository? I think this could be a great approach to open, transparent discussion and resolutions in future iterations of this dataset. If you would like to use the repository as a user contribution portal then I would recommend: 1) adding a section to the readme on how to contribute; 2) adding an issue template to guide users in writing their contributions; and 3) adding a repo action to check the compatibility of user contributions (e.g. ensuring the submission is a .csv with all essential fields included).

The discussions are not openly available, simply because they largely occurred in an unstructured manner (e.g. by email or whatever form of communication regional teams preferred). One example of such a discussion is visible as issue #4 on the GitHub repository (https://github.com/GLIMS-RGI/lake\_terminating/issues/4). Considering that many contributors now (and possibly also in future) are not familiar with how github works and git-literacy shouldn't necessarily be a requirement to contribute on this topic, we also believe that requirements for exchange on these issues or future contributions should not be too stringent. However, we agree that easing the submission process for potential future contributors should be attempted at this stage. We have therefore included a section to the ReadMe on how to contribute, including an example template for users. We have now included a GitHub Action to check contributions (PR#15), and have included additional instructions in the Contributing section of the README.

Line 228: The .gpkg information should be included when describing the format and contents of the .csv tables (Line 123-127), rather than at the end of the manuscript. In addition, the geographic projection (OGC:CRS84) and field descriptions (i.e. fid, IDs, aut\_trm, lak\_lvl, image\_d, imag\_dt, invntr\_, cntrbtr, notes) of the gpkg files should also be included. Perhaps the field description names in the .gpkg files could be incorporated into Table 2.

Thanks for pointing this out - the attributes in the gpkgs are actually the same as described in Table 2, but the names were corrupted in the process of preparing the packages. This is now updated.

Line 254-256: Normally ESSD publications require a section, or some comment, and how this dataset could be used in future work. I think a couple of comments could easily be added to the Conclusions here, tying back to the relevant literature highlighted in the introduction.

We have expanded on the very last sentence of the Conclusions where we noted the relevance of the dataset for future studies, noting its suitability for scaling in-situ insights.

**References**

Camassa, R. et al. (2023) Extreme seasonal water-level changes and hydraulic modeling of deep, high-altitude, glacial-carved, Himalayan lakes. Sci Rep 13, 11705. https://doi.org/10.1038/s41598-023-37667-z

Steffen, T. et al. (2022) Volume, evolution, and sedimentation of future glacier lakes in Switzerland over the 21st century, Earth Surf. Dynam., 10, 723–741, https://doi.org/10.5194/esurf-10-723-2022

Veh, G. et al. (2025) Progressively smaller glacier lake outburst floods despite worldwide growth in lake area. Nat Water 3, 271–283. <a href="https://doi.org/10.1038/s44221-025-00388-w">https://doi.org/10.1038/s44221-025-00388-w</a>

---

## Author Comment (AC2)

Response to comments on "Global mapping of lake-terminating glaciers" (essd-2025-315)

**Reviewer 3**

This study presents a global inventory of lake-terminating glaciers based on the RGI 7.0 glacier inventory and published datasets of glacial lakes. The key approach involves determining the relationship between glaciers and glacial lakes (direct contact or not) in different cases, considering the ambiguity in the published datasets, which map glaciers/lakes based on images with varying time stamps that may differ by several years or over a decade. The authors adopted a manual interpretation strategy, with clear guidance on different categories of glacier-lake relationships, each implying a different level of confidence on the impact of lake water on glacier dynamics. The manuscript is well-written and logically clear, except for a few instances of inappropriate naming (e.g., lake-level) and ambiguous implementation of different classifications. I believe that this dataset with consistent quality control benefits future investigation and modelling of glacier dynamics (velocity changes, mass balance) in response to climate warming.

Thank you for the in-depth review, we respond to all concerns with the proposed changes individually below. Please note that following suggested changes from other reviewers, some of the basic terminology and ordering of 'categories' (previously 'levels') have now changed.

**Major comments**

My main concern lies in the ambiguous definition of lake-terminating glaciers and how the current classification represents in the temporal span. If the lake-terminating glaciers refer to glacier terminuses that are in direct contact with glacial lake water, then the manual inspection is to rule out glaciers with supraglacial lakes and glacial lakes surrounding glaciers in the lateral zones. However, Figure 2d (lake-level 2) appears to include glaciers that are not directly in contact with the glacier terminus. It seems that lake-level 3 are not regarded as lake-terminating glaciers in the results and analysis (Line 135-137). In addition, it is also interesting to see how the data products of this study differ from a simple classification method, e.g., flagging lake-terminating glaciers by simply overlapping the buffering of glacier outlines with lake outlines. Such a comparison can help understand how the expert inspections implemented in this study improve the classifications.

The concerns regarding the introduction of some sort of ambiguity is understandable and we agree that allowing 'expert judgement' to make the call introduces some grey zone where definitions are not as clear cut as they would be in an automated approach. To be clear, the automated approach was originally favoured, but evaluating the available data (lake inventories, consistency across regions) as well as the difficulty in making automated judgements (What to do about a lake that is visibly in contact with ice but with an outline that does not intersect with the glacier outline, both of which in turn are at times not precise? When is a lake at the terminus and when is it just located along the glacier margin?) led us to converge on the more manual approach. To underline the difficulty, we show below a comparison between the approach taken in the study, vs a much more traceable automated approach, namely identifying all glaciers from the RGI as lake-terminating that intersect with

a lake from three global inventories (Shugar et al. 2020; Zhang et al. 2024; Song et al. 2025, Table 1, Figure 1, Figure 2). The results highlight a number of issues:

- a) As we show in the manuscript, the differences in lake number and area between the global inventories is large (Table 1, Figure 2), begging the question, which one to rely on as adequate.
- b) In many regions the numbers of lake-terminating glaciers identified by the automated approach comes closer to the numbers we find in our study, when we move from a very crude 'intersect glacier with all lakes' approach to the more refined 'within buffer around terminus and 100 m away from glacier outline' approach, but this is not always the case and it rarely brings the number to a match (Table 1, Figure 2 and 3).
- c) The simple intersection between lake and glacier does provide numbers of in similar orders of magnitude in some regions but not others (Figure 3, Table 1) but can not tell us whether the lake is at the terminus.
- d) While applying a buffer is easily done, adding 100 m around glacier termini increases the number of glaciers intersecting with lakes considerably in most regions (Table 1) but there is no way of telling whether 100 m is adequate at a global scale. We can show that when the buffer is increased, we find a much larger agreement rate (Figure 1), but also false positives increase rapidly (Figure 4), which then puts us back in the situation of having to evaluate manually.
- e) The difference between automatically and manually flagged glaciers is always large, and not consistent. While we do not want to argue that the lake inventories are inadequate indeed we use them as an aid the large differences across all regions (mapped by different contributors) suggests that an automated approach alone does not adequately address the issue.
- f) The approaches also disagree on many glaciers, with many automated lake termini not being recognized as such by our approach and vice versa (see Table 1) for certain regions, while in others it could work, but the issue of false positives remains.
- g) The automated approach would not allow us to differentiate between different categories as we are able to do with an expert informed approach.

As we note in the manuscript, we do not want to suggest that the expert judgement based approach is perfect, but we argue that with the current datasets and methods available, it seems prudent to provide a baseline dataset based on this expert judgement. Future improvements in data analysis, may further move the classification towards a more reproducible automated method.

We now suggest to include a more detailed argument for this decision in the Methods section, and provide a number of figures that outline the potential alternative approach with global and regional inventories as an Appendix.

Table 1: Number of glaciers identified as lake-terminating by the approach followed in this study (*This study*, including all categories 2 and 3), an automated approach intersecting the inventory by Shugar et al. (2020) (time period 2000 +4 therein) with all glaciers in the RGI7 (*Shugar2020*), an approach where we place a buffer of 100 m around the glacier before intersecting (*Shugar2020\**), and a version where we only do this within a radius of 1 km around the terminus position, according to RGI7 (Shugar2020\*\*). This last column corresponds to the approach taken for the following figures, where then different buffer sizes

are tested. In brackets are true positives in comparison to the approach taken in this study. The same data are shown in comparison to the data from Zhang et al. (2024) and Song et al. (2025). Note that the datasets of Shugar2020 and Song2025 have no data in regions 7 and 19.

| RGI    | This  | Shugar       | Shugar       | Shugar       | Zhang        | Zhang        | Zhang        | Song2        | Song2         | Song2         |
|--------|-------|--------------|--------------|--------------|--------------|--------------|--------------|--------------|---------------|---------------|
| Region | study | 2020         | 2020*        | 2020**       | 2024         | 2024*        | 2024**       | 025          | 025*          | 025**         |
|        |       |              |              |              |              |              |              |              |               |               |
| 1      | 187   | 209          | 193          | 180          | 332          | 387          | 362          | 935          | 998           | 985           |
|        |       | (82)         | (95)         | (89)         | (55)         | (64)         | (52)         | (176)        | (175)         | (175)         |
| 2      | 352   | 127          | 233
(104) | 232          | 408          | 742          | 733          | 1060         | 1381          | 1361          |
|        |       | (67)         | , ,          | (104)        | (105)        | (149)        | (148)        | (307)        | (334)         | (332)         |
| 3      | 184   | 138
(26)  | 26 (17)      | 23 (14)      | 634
(136) | 242
(92)  | 221
(85)  | 324
(79)  | 114
(45)   | 106
(42)   |
| 4      | 184   | 346          | 342          | 323          | 424          | 539          | 498          | 1120         | 1201          | 1147          |
| _      | 104   | (100)        | (95)         | (92)         | (88)         | (111)        | (102)        | (167)        | (150)         | (144)         |
| 5      | 294   | 376          | 414          | 400          | 913          | 886          | 824          | 936          | 945           | 894           |
|        |       | (111)        | (115)        | (112)        | (169)        | (174)        | (159)        | (184)        | (180)         | (173)         |
| 6      | 14    | 23 (8)       | 13 (7)       | 10 (6)       | 25 (10)      | 17 (11)      | 17 (11)      | 90 (14)      | 58 (14)       | 57 (14)       |
| 7      | 60    | 0            | 0            | 0            | 75 (18)      | 47 (18)      | 44 (17)      | 0            | 0             | 0             |
| 8      | 227   | 131          | 208          | 204          | 98 (72)      | 201          | 197          | 395          | 492           | 481           |
|        |       | (57)         | (73)         | (72)         |              | (111)        | (110)        | (190)        | (199)         | (198)         |
| 9      | 32    | 132
(20)  | 90 (21)      | 83 (21)      | 100
(12)  | 56 (19)      | 44 (14)      | 115
(23)  | 63 (25)       | 59 (25)       |
| 10     | 69    | 25 (9)       | 31 (10)      | 31 (10)      | 62 (21)      | 121          | 121          | 232          | 295           | 294           |
|        |       |              |              |              |              | (41)         | (41)         | (53)         | (59)          | (59)          |
| 11     | 66    | 8 (5)        | 13 (7)       | 13 (7)       | 7 (4)        | 21 (9)       | 21 (9)       | 87 (42)      | 113
(48)   | 112
(47)   |
| 12     | 11    | 5 (2)        | 5 (3)        | 5 (3)        | 2 (1)        | 13 (4)       | 13 (4)       | 60 (9)       | 74 (10)       | 74 (10)       |
| 13     | 715   | 194          | 256          | 255          | 307          | 856          | 851          | 1575         | 2092          | 2079          |
|        |       | (108)        | (143)        | (143)        | (135)        | (338)        | (335)        | (433)        | (529)         | (527)         |
| 14     | 337   | 52 (33)      | 76 (62)      | 76 (62)      | 159          | 382          | 376          | 574          | 767           | 763           |
|        |       |              |              |              | (69)         | (193)        | (193)        | (230)        | (283)         | (283)         |
| 15     | 420   | 180
(140) | 299
(191) | 296
(191) | 219
(112) | 555
(252) | 546
(251) | 788
(294) | 1068
(360) | 1048
(358) |
|        |       | (140)        | (181)        | (181)        | (112)        | (232)        | (201)        | (234)        | (300)         | (336)         |

| 16 | 72  | 43 (24)      | 57 (28)      | 55 (28)      | 29 (13)      | 95 (28)      | 94 (28)      | 237
(54)  | 302
(64)   | 297
(64)   |
|----|-----|--------------|--------------|--------------|--------------|--------------|--------------|--------------|---------------|---------------|
| 17 | 575 | 143
(105) | 212
(135) | 209
(132) | 433
(196) | 780
(280) | 770
(273) | 884
(402) | 1272
(450) | 1261
(447) |
| 18 | 15  | 13 (9)       | 15 (9)       | 15 (9)       | 7 (3)        | 10 (5)       | 9 (4)        | 360
(13)  | 48 (12)       | 48 (12)       |
| 19 | 22  | 0            | 0            | 0            | 26           | 30 (15)      | 28 (13)      | 0            | 0             | 0             |

Figure 1: Agreement rate between the manual mapping done in this study, against an automated approach using three different global lake inventories. For each RGI glacier, we intersected a buffer of 0, 100, 500, and 1000 m around the terminus coordinates with the lake inventories. The agreement rate is the number of glaciers where both the automated approach and the manual approach agreed, divided by the number of lake-terminating glaciers mapped manually. However this also leads to more false positives (See Figure 3).

Figure 2: Agreement rate between the manual mapping done in this study, against an automated approach using three different global lake inventories. For each RGI glacier, only for the 0 km buffer. The agreement rate is the number of glaciers where both the automated approach and the manual approach agreed, divided by the number of lake-terminating glaciers mapped manually. However this also leads to more false positives (See Figure 3).

Figure 3: Number of lake-terminating glaciers identified using a buffer of 0 km (i.e., lakes intersect terminus). Black lines show what was manually identified in each region.

Figure 4: False positive rate for regional inventories, grouped by buffer size in meters. False positive rate is defined as the number of glaciers that were flagged as lake-terminating by the automated approach that were not flagged by the manual approach, divided by the number of glaciers flagged by the manual approach.

A second concern is about the stability of the current classification of lake-terminating glaciers, given the dynamics of glaciers and glacial lakes, particularly in certain regions such as High Mountain Asia, which show high increases in the number and area of glacial lakes in recent decades (Shugar et al., 2020; Zhang et al., 2024). I understand the alignment of the reference year 2000 for consistency; however, the 'imperfect' nature of current glacier and lake mapping, e.g., time consistency and resolution, may imply that such a classification needs to be representative over a time span, e.g., 1999-2005. This is also necessary for future studies to examine the different behaviors of lake-terminating glaciers over a relatively long time span (otherwise, we would need to determine the glacier type case by case, e.g., when the glacier becomes connected with lake water and when it detaches from lakes). From this perspective, I support the inclusion of some cases (e.g., Figure 3a) as lake-terminating glaciers, or by referencing lake outlines/images from nearby years. At least, the authors need to discuss the potential applications and the cautions need to be paid with the current classifications.

We agree that the dataset needs to be used with care, and the link to the time period around 2000 is important. We have discussed this with the analysis around Figure 8. We now additionally emphasize this in the second to last paragraph of the Conclusions as well.

**Specific comments**

I have very few additional specific comments, as the manuscript is well-written, and some details have been pointed out in other comments.

Title: A title such as "A global inventory of lake-terminating glaciers" may be more appropriate as the work is essentially a classification of glacier types based on existing mapping results.

We would prefer to stick to the current title as 'a global inventory of [...] glaciers' would suggest we are providing a new inventory here, while we are actually working with an existing one (RGI7).

Line 132-133: marine-terminating glaciers are given a term\_type value of 1 or 3? It is mentioned that they are assigned as type 0 in Lines 106-107.

The attribute **term\_type** is from RGI7 and pertains to the type of terminus (not the specific category of lake termini). Glaciers that end in the ocean are assigned the category of 0 if they do not also end in a lake.

Line 262-264: This implies that we could rely on the current classifications for assessing the long-term glacier mass balance of lake-terminating glaciers and others. However, the direct contact of glacial lake water with the glacier terminus can change rapidly, given the strong glacier terminus retreat and expansion of glacial lakes, which have been widely reported in the mountainous glacierized regions in the past decades. A single reference year (e.g., 2000) of the classification does not seem to be sufficient.

In a potential modelling study the inventory should of course not be used as a static fact, but could be used as an initial known state. We have now clarified this with the addition of 'as an initial state'.

**References:**

Shugar, D. H., Burr, A., Haritashya, U. K., Kargel, J. S., Watson, C. S., Kennedy, M. C., Bevington, A. R., Betts, R. A., Harrison, S., & Strattman, K. (2020). Rapid worldwide growth of glacial lakes since 1990. *Nature Climate Change*, *10*(10), 939–945. <a href="https://doi.org/10.1038/s41558-020-0855-4">https://doi.org/10.1038/s41558-020-0855-4</a>

Song, C., Fan, C., Ma, J. *et al.* A spatially constrained remote sensing-based inventory of glacial lakes worldwide. *Sci Data* **12**, 464 (2025). https://doi.org/10.1038/s41597-025-04809-z

Zhang, T., Wang, W. & An, B. Heterogeneous changes in global glacial lakes under coupled climate warming and glacier thinning. *Commun Earth Environ* **5**, 374 (2024). https://doi.org/10.1038/s43247-024-01544-y

---

## Author Comment (AC3)

Response to comments on "Global mapping of lake-terminating glaciers" (essd-2025-315)

**Reviewer 2 (Johnny Ryan)**

This manuscript describes an approach to assign a "lake-terminating relevance level" to all glaciers in the Randolph Glacier Inventory (RGI). The study mainly uses published proglacial lake inventories to search for glaciers that are likely to be lake-terminating. The authors then use expert judgement to assign individual glaciers a level (between 0 and 3). I think that this is a great idea that will be a valuable addition to the RGI. Generally, I found the manuscript to be clearly presented. However, I thought that the execution could be improved. My main concern is that the requirements of the categories seem be used selectively (e.g. category definitions are also not strictly adhered to, reliance of single satellite images to infer "visual impact" of lake). Although the authors do a good job of showing general agreement between multiple experts, I think that this source of ambiguity will make the dataset difficult to update for another time period. Overall, I believe that this dataset will definitely be a useful contribution to ESSD but I encourage the authors to revisit the definitions and implementation of their categorization scheme.

Thank you for the in-depth review. We are grateful for the careful reading and reflection on the methods especially, which have led us to evaluate the data further. We respond to all concerns with the proposed changes individually below.

**Major comment**

There is a mismatch in the category definitions and the implementation of the classification. For example, there is a strict requirement that the ice-lake interface must be >50% (Level 1), <50% and >10% (Level 2), and

a little more transparent about the subjectivity so that others can understand the strengths and weaknesses of the classification approach.

We appreciate your concern regarding the subjectivity introduced through manual approaches. Our preferred approach would have been to make this purely objective from the start, with available lake inventories and an automated intersection process. Indeed this is how we set out, only to quickly come to the conclusion that with the available data this would leave us with an unsatisfying result. To underline this point we have carried out a string of additional analyses, comparing a potentially more 'automated' approach with the more 'supervised' path we have chosen here, which we present below. We feel that it goes beyond the space in this manuscript to include all of this discussion, but we include some of the takeaways in the Methods description as well as the Results. We agree that language can be improved to reduce ambiguity (or as you suggest emphasize subjectivity when there is) and we have followed this advice in the description of the categories (previously called level, changed following comments by reviewer 1).

To further evaluate what a more traceable and automated approach would look like we have now spent some time to play potential scenarios and evaluate results against what we have found in this study. This includes investigating automated intersections in various ways with available inventories, the results of which we propose to add as an Appendix to the Methods section and which we show below in response to the specific questions. While we agree that it would be possible to follow such an automated approach, we think the results visualize why the - subjective - expert judgement remains an important component. We also hope that making available multiple lake categories that may hold some potential for argument whether a glacier belong in one or the other category, provides users to make a choice - if you are interested in all glaciers that have an interaction with lake water, all three categories may be brought together, if you are only interested in glaciers to e.g. apply complex numerical calving studies, just Category 3 (what was Level 1) may be in order but users could also take Category 3 and 2 and make individual choices on which of the rather limited number in total they wish to include.

We detail this additional analysis in response to one of the specific questions below.

**Specific comments**

L12: Could be a little more specific about "contribute to glacier velocity" to match the directional intent of the second statement in the sentence?

Expanded to 'Their presence has been shown to result in increased glacier velocities and therefore drive...'

L15-16: I'm not sure cherry-picking one study that modeled one glacier in New Zealand is very compelling evidence for this statement. I don't doubt that this is true but are there not observations from a sample of glaciers that could better support this claim?

Thanks for the suggestion. We have now decided to group the findings from this study with the previous, but making clearer the distinction in results (whether velocity, grounding line or buoyancy changes).

Their presence has been shown to result in increased glacier velocities (Pronk et al., 2021; Minowa et al., 2023; Baurley et al., 2020), unstable termini due to buoyancy (Boyce et al., 2007; Trüssel et al., 2013; Main et al., 2023) or accelerated grounding line recession (Sutherland et al., 2020) individually or in combination driving dynamic thinning (Tsutaki et al., 2019; Larsen et al., 2015; King et al., 2019).

L19-22: It would be helpful to provide some more evidence for the statement about calving (L19). I see that the next sentence mentions 24 Gt a-1 from Patagonia but frontal ablation could all be submarine melt. It would also be useful to provide some context for magnitude of 24 Gt a-1 relative to total mass loss or something.

We have rephrased this section, which should accommodate the concern raised.

Calving into lakes is an important driver of glacier mass loss (Warren and Aniya, 1999), but remains poorly constrained with relatively few observations. Retreat rates, that can not always differentiate between subaqueous melt and calving, were found to be between 20 to 70 m a-1 in the Himalaya (Watson et al., 2020; Pratap et al., 2025) and 800 m a-1 in Patagonia. Minowa et al. (2021) found that lake-terminating glaciers in Patagonia collectively lost  $\sim$  24 Gt a-1 through frontal ablation on average over 2000 - 2019

In the process we have removed the Sakai et al, (2009) citation, since it largely pertains to supraglacial lakes and statements are backed up sufficiently by the remaining studies.

L20: What is meant by "scattered"

Removed with above change.

L24: Consider summarizing this paragraph with a sentence about the importance of lake-terminating glaciers.

We suggest a final sentence here: "This wide range of studies, if limited by limited in-situ observations, suggests that lake-terminating glaciers exhibit distinct properties that are crucial to consider, especially in light of regional or global investigations of glacier dynamics."

L26: Not sure what "regional assessments and case studies" is referring to here. Contribution of lake-terminating glaciers to mass loss? Mass loss from frontal ablation at the ice-lake interface? Underestimation of mass loss from lake-terminating glaciers?

In line with the above addition, we now shorten this to 'However, insights from these studies ...'

L28-29: This statement is at odds to the first sentence of the paragraph.

The first statement is about an assessment of whether glaciers are lake-terminating; the statement at lines 28-29 is about inventories of glacial lakes.

L28-31: The difference implies that 1) one study has large errors, 2) both studies have large errors, or 3) there was enormous growth in lake numbers and volume between 2018. Given that (3) is unlikely, I would encourage the authors to just come out and say that they suspect errors in these datasets, perhaps commenting on some possible causes.

It's beyond this study to provide a detailed assessment of where the issues lie within both studies - from visual inspection, it is obvious that many lakes were missed in both, likely due to the automated approach understandably taken at this scale and the resolution of the imagery employed. We now add this as a possible explanation after introducing these two datasets.

L31: "Both" instead of "All"?

**Changed.**

L35-36: OK so some of the glaciers do have this information? I think the authors should describe the number of glaciers or regions which have (or don't have) this information to more clearly motivate this study.

RGI6.0 had information about lake-terminating status for three regions: Alaska, Southern Andes, and Antarctica, but this has not yet been included in RGI7. To clarify this point, we have modified this sentence.

L38-39: Recommend adding some more background to this paragraph. For example, why was 2000 chosen as the target year? How many lake-terminating glaciers were identified? How does identifying lake-terminating glaciers improve the RGI?

2000 was chosen as the target year because that is the target year for RGI7. See, for example: https://www.glims.org/rgi\_user\_guide/01\_introduction.html#what-is-the-rgi

As for the improvement to RGI, we think this should now also be addressed with the concluding sentence of the first paragraph (the use for regional/global modelling when relying on the general attributes stored within RGI).

We have added the numbers of lake termini that were previously mapped in the respective areas to the manuscript.

L57: How were lakes

This statement is not about the percent of ice-lake contact but about where the lake is located. We revised the text to make this clearer.

There are a few instances in the Canadian Arctic of an ice-dammed lakes away from the terminus that resulted in the glacier being flagged as lake\_category = 1 (in the new system). This occurred in a limited number of cases, and served essentially as flags for potential sites of interest for future research & dataset refinement. These limited number of ice-dammed lakes away from the terminus wouldn't affect our final results, in which glaciers are considered lake -terminating if they have a lake category of 2 or 3.

L116-119: It's not clear how prior labelling reduces future subjectivity. Future efforts will have to use just as many subjective judgement calls.

A fair point. We have revised this to state that using this framework alongside multi-temporal glacier inventories will enable investigation of how these glaciers (and lakes) develop over time.

L122: "following a simple structure" of what?

Updated to read "following the structure indicated in Table 2."

L130-131: So some glaciers in RGI7 are already classified as lake-terminating? If this is so, then that should be outlined in the introduction e.g. for which regions, how many glaciers etc.

No glaciers have been classified as lake-terminating in RGI7, as the only glaciers with **term\_type** set are marine-terminating glaciers (see, e.g., <a href="https://www.glims.org/rgi\_user\_guide/products/glacier\_product.html#terminus-type">https://www.glims.org/rgi\_user\_guide/products/glacier\_product.html#terminus-type</a> and

https://www.glims.org/rgi\_user\_guide/06\_dataset\_summary.html#global-attributes-statistics). RGI 6.0 did have this attribute set, but only for some regions (Alaska, Southern Andes, and Antarctica.

L137-138: The definitions seem to have been discarded if Level 3 now represents "only some adjacent water bodies, but without significant interaction with the glacier ice". Level 3 has a, somewhat specific, requirement for

Figure 1: Number of lake-terminating glaciers identified using a buffer of 0 km (i.e., lakes intersect terminus). Black lines show what was manually identified in each region.

Figure 2: Agreement rate between the manual mapping done in this study, against an automated approach using three different global lake inventories. For each RGI glacier, we intersected a buffer of 0, 100, 500, and 1000 m around the terminus coordinates with the lake inventories. The agreement rate is the number of glaciers where both the automated approach and the manual approach agreed, divided by the number of lake-terminating glaciers mapped manually. However this also leads to more false positives (See Figure 3).

Figure 3: False positive rate for regional inventories, grouped by buffer size in meters. False positive rate is defined as the number of glaciers that were flagged as lake-terminating by the automated approach that were not flagged by the manual approach, divided by the number of glaciers flagged by the manual approach.

Table 1: Number of glaciers identified as lake-terminating by the approach followed in this study (*This study*, including all categories 2 and 3), an automated approach intersecting the inventory by Shugar et al. (2020) (time period 2000 +4 therein) with all glaciers in the RGI7 (*Shugar2020*), an approach where we place a buffer of 100 m around the glacier before intersecting (*Shugar2020\**), and a version where we only do this within a radius of 1 km around the terminus position, according to RGI7 (Shugar2020\*\*). This last column corresponds to the approach taken for the following figures, where then different buffer sizes are tested. In brackets are true positives in comparison to the approach taken in this study. The same data are shown in comparison to the data from Zhang et al. (2024) and Song et al. (2025). Note that the datasets of Shugar2020 and Song2025 have no data in regions 7 and 19.

| RGI    | This  | Shugar      | Shugar      | Shugar      | Zhang       | Zhang       | Zhang       | Song2        | Song2        | Song2        |
|--------|-------|-------------|-------------|-------------|-------------|-------------|-------------|--------------|--------------|--------------|
| Region | study | 2020        | 2020*       | 2020**      | 2024        | 2024*       | 2024**      | 025          | 025*         | 025**        |
| 1      | 187   | 209
(82) | 193
(95) | 180
(89) | 332
(55) | 387
(64) | 362
(52) | 935
(176) | 998
(175) | 985
(175) |

| 2  | 352 | 127
(67)  | 233
(104) | 232
(104) | 408
(105) | 742
(149) | 733
(148) | 1060
(307) | 1381
(334) | 1361
(332) |
|----|-----|--------------|--------------|--------------|--------------|--------------|--------------|---------------|---------------|---------------|
| 3  | 184 | 138
(26)  | 26 (17)      | 23 (14)      | 634
(136) | 242
(92)  | 221
(85)  | 324
(79)   | 114
(45)   | 106
(42)   |
| 4  | 184 | 346
(100) | 342
(95)  | 323
(92)  | 424
(88)  | 539
(111) | 498
(102) | 1120
(167) | 1201
(150) | 1147
(144) |
| 5  | 294 | 376
(111) | 414
(115) | 400
(112) | 913
(169) | 886
(174) | 824
(159) | 936
(184)  | 945
(180)  | 894
(173)  |
| 6  | 14  | 23 (8)       | 13 (7)       | 10 (6)       | 25 (10)      | 17 (11)      | 17 (11)      | 90 (14)       | 58 (14)       | 57 (14)       |
| 7  | 60  | 0            | 0            | 0            | 75 (18)      | 47 (18)      | 44 (17)      | 0             | 0             | 0             |
| 8  | 227 | 131
(57)  | 208
(73)  | 204
(72)  | 98 (72)      | 201
(111) | 197
(110) | 395
(190)  | 492
(199)  | 481
(198)  |
| 9  | 32  | 132
(20)  | 90 (21)      | 83 (21)      | 100
(12)  | 56 (19)      | 44 (14)      | 115
(23)   | 63 (25)       | 59 (25)       |
| 10 | 69  | 25 (9)       | 31 (10)      | 31 (10)      | 62 (21)      | 121
(41)  | 121
(41)  | 232
(53)   | 295
(59)   | 294
(59)   |
| 11 | 66  | 8 (5)        | 13 (7)       | 13 (7)       | 7 (4)        | 21 (9)       | 21 (9)       | 87 (42)       | 113
(48)   | 112
(47)   |
| 12 | 11  | 5 (2)        | 5 (3)        | 5 (3)        | 2 (1)        | 13 (4)       | 13 (4)       | 60 (9)        | 74 (10)       | 74 (10)       |
| 13 | 715 | 194
(108) | 256
(143) | 255
(143) | 307
(135) | 856
(338) | 851
(335) | 1575
(433) | 2092
(529) | 2079
(527) |
| 14 | 337 | 52 (33)      | 76 (62)      | 76 (62)      | 159
(69)  | 382
(193) | 376
(193) | 574
(230)  | 767
(283)  | 763
(283)  |
| 15 | 420 | 180
(140) | 299
(191) | 296
(191) | 219
(112) | 555
(252) | 546
(251) | 788
(294)  | 1068
(360) | 1048
(358) |
| 16 | 72  | 43 (24)      | 57 (28)      | 55 (28)      | 29 (13)      | 95 (28)      | 94 (28)      | 237
(54)   | 302
(64)   | 297
(64)   |
| 17 | 575 | 143
(105) | 212
(135) | 209
(132) | 433
(196) | 780
(280) | 770
(273) | 884
(402)  | 1272
(450) | 1261
(447) |
| 18 | 15  | 13 (9)       | 15 (9)       | 15 (9)       | 7 (3)        | 10 (5)       | 9 (4)        | 360
(13)   | 48 (12)       | 48 (12)       |
| 19 | 22  | 0            | 0            | 0            | 26           | 30 (15)      | 28 (13)      | 0             | 0             | 0             |

Table 1 provides a summary of a number of approaches for all datasets, starting from the simple intersection of glaciers with lakes (no \*), to an approach with a buffer (\*) and finally the most zoomed in version we apply a buffer around the glacier to intersect with potential lakes only within a 1 km radius of the glacier terminus (\*\*, see also Figure 2). It is evident that these different approaches allow us to narrow down our selection but do not leave us with a finally satisfying dataset. We therefore propose the following edits

- a) As suggested above, we provide clearer language on the subjectivity of the approach and make sure that it remains consistent for all categories throughout the manuscript.
- b) We add additional text to the Methods section summarizing the automated potssibilities, their advantages and shortcoming, which led us to our final choice.
- c) We provide Figures summarising the results from exploring all datasets and potential buffering approaches in an Appendix.

L186: Which results? What is meant by "relative importance"? Submarine melt? Ice flow? Mass loss?

Updated to 'our inventory' and 'relative abundance'.

L189-190: Long, wordy sentence, consider revising.

**Reworded.**

L190-191: Explain whether this is good or bad.

Now added a line on this being a satisfactory match, considering that morphology change would mostly happen at longer time scales.

L204-205: This begs the question: why didn't the authors use the same images as the ones used for the glacier outlines?

This is a fair point, and one that was considered early in the process. However, because of the additional investigation required to determine what original imagery was used for each glacier, alongside the potentially large amount of imagery that would need to be downloaded and shared among contributors, we determined that it was most straightforward to select a smaller range of primarily Landsat imagery.

L242: I've made a similar point before but I'm struggling with the claim that the dynamics of all of these glaciers (Level 1 and 2) are significantly altered by lakes. A single multispectral image cannot provide that much information about dynamics.

We have now removed the explicit links to dynamics, but only refer to these lakes having obvious ice-water interfaces.

L250: If Level 1 and 2 are collectively termed "lake-terminating" and Level 0 and 3 are "land-terminating", then this reduces the need for four categories. One solution would be to use a smaller number of categories.

We disagree. Here, we are combining the different categories to allow for an update for the RGI7.1 attributes, which does not differentiate between the level of contact between glacier and lake. We believe that it is still useful to use the framework we have developed here to allow for a finer-grained approach to investigating the interaction between glaciers and lakes. This still allows users to eventually only consider a binary land-terminating vs lake-terminating distinction but provides a baseline for future investigations if this may change or produce relevant information for studies that may consider lakes relevant even if not

L255-256: The number of lakes does not necessarily imply that they have an important role. Consider adding some citations to studies that have demonstrated this.

We have rephrased this, highlighting the presence of two regional studies assessing their role and suggesting that this dataset can provide the basis for investigating whether these patterns hold true elsewhere as well as at the global scale.

L259-260: I found this to be a little disappointing. The authors make a big deal about previous studies not determining whether lakes were in direct contact with glaciers (180-187). They then explicitly use "direct contact" as a requirement in their categorization. But only now do we find out that the analysis does not "does not consider the actual number of lakes in contact with an individual glacier or the length of the ice-water interface". I think this should have been one of the primary goals of the present analysis.

This is a fair point (and one that we also discussed in the early stages), but we felt that asking participants to do the additional work of manually digitizing the ice-water interface, or digitize individual lakes, was more than required for the primary goal of identifying lake-terminating glaciers around the world. We rather see this as a first step that would eventually allow us to further investigate what is happening for the subset of glaciers with lake termini, which again requires additional discussion of methods, that, we felt, would go beyond the scope of one manuscript.

**References:**

Kaufman, D. S., & Manley, W. F. (2004). Pleistocene maximum and Late Wisconsinan glacier extents across Alaska, USA. In Developments in quaternary sciences (Vol. 2, pp. 9-27). Elsevier.

Péwé, T. L. (1975). Quaternary geology of Alaska (Vol. 835). US Government Printing Office.

Shugar, D. H., Burr, A., Haritashya, U. K., Kargel, J. S., Watson, C. S., Kennedy, M. C., Bevington, A. R., Betts, R. A., Harrison, S., & Strattman, K. (2020). Rapid worldwide growth of glacial lakes since 1990. *Nature Climate Change*, *10*(10), 939–945. <a href="https://doi.org/10.1038/s41558-020-0855-4">https://doi.org/10.1038/s41558-020-0855-4</a>

Song, C., Fan, C., Ma, J. *et al.* A spatially constrained remote sensing-based inventory of glacial lakes worldwide. *Sci Data* **12**, 464 (2025). https://doi.org/10.1038/s41597-025-04809-z

Vamvaka, A., Pross, J., Monien, P., Piepjohn, K., Estrada, S., Lisker, F., & Spiegel, C. (2019). Exhuming the top end of North America: episodic evolution of the Eurekan belt and its potential relationships to North Atlantic plate tectonics and Arctic climate change. Tectonics, 38(12), 4207-4228.

Zhang, T., Wang, W. & An, B. Heterogeneous changes in global glacial lakes under coupled climate warming and glacier thinning. *Commun Earth Environ* **5**, 374 (2024). https://doi.org/10.1038/s43247-024-01544-y